# Effect of High Versus Low Carbohydrate Intake in the Morning on Glycemic Variability and Glycemic Control Measured by Continuous Blood Glucose Monitoring in Women with Gestational Diabetes Mellitus—A Randomized Crossover Study

**DOI:** 10.3390/nu12020475

**Published:** 2020-02-13

**Authors:** Louise Rasmussen, Maria Lund Christensen, Charlotte Wolff Poulsen, Charlotte Rud, Alexander Sidelmann Christensen, Jens Rikardt Andersen, Ulla Kampmann, Per Glud Ovesen

**Affiliations:** 1Department of Nutrition, Exercise and Sports, University of Copenhagen, Rolighedsvej 26, 1958 Frederiksberg C, Copenhagen, Denmark; louiseras90@gmail.com (L.R.); mclund89@gmail.com (M.L.C.); jra@nexs.ku.dk (J.R.A.); 2Department of Obstetrics and Gynecology, Aarhus University Hospital, Palle Juul-Jensens Boulevard 99, 8200 Aarhus N, Denmark; charpoul@rm.dk (C.W.P.); per.ovesen@clin.au.dk (P.G.O.); 3Department of Hepatology and Gastroenterology, Aarhus University Hospital, Palle Juul-Jensens Boulevard 99, 8200 Aarhus N, Denmark; charru@rm.dk; 4Steno Diabetes Center Copenhagen, Niels Steensens Vej 2-4, 2820 Gentofte, Denmark; alexander.sidelmann.christensen@regionh.dk; 5Clinic for Clinical Metabolic Research, Herlev and Gentofte Hospital, University of Copenhagen, Gentofte Hospitalsvej 7, 3. Sal, 2800 Hellerup, Denmark; 6Steno Diabetes Center Aarhus, Aarhus University Hospital, Hedeager 3, 8200 Aarhus N, Denmark

**Keywords:** glycemic variability, carbohydrate distribution, gestational diabetes mellitus, mean amplitude of glucose, breakfast diet

## Abstract

Carbohydrate is the macronutrient that has the greatest impact on blood glucose response. Limited data are available on how carbohydrate distribution throughout the day affects blood glucose in women with gestational diabetes mellitus (GDM). We aimed to assess how a high-carbohydrate morning-intake (HCM) versus a low-carbohydrate-morning-intake (LCM), affect glycemic variability and glucose control. In this randomized crossover study continuous glucose monitoring (CGM) was performed in 12 women with diet treated GDM (75 g, 2-h OGTT ≥ 8.5 mmol/L), who went through 2 × 3 days of HCM and LCM. A within-subject-analysis showed a significantly higher mean amplitude of glucose excursions (MAGE) (0.7 mmol/L, *p* = 0.004) and coefficient of variation (CV) (5.1%, *p* = 0.01) when comparing HCM with LCM, whereas a significantly lower mean glucose (MG) (−0.3 mmol/L, *p* = 0.002) and fasting blood glucose (FBG) were found (−0.4 mmol/L, *p* = 0.01) on the HCM diet compared to the LCM diet. In addition, insulin resistance, expressed as Homeostatic Model Assessment for Insulin Resistance (HOMA-IR), decreased significantly during HCM. Results indicate that a carbohydrate distribution of 50% in the morning favors lower blood glucose and improvement in insulin sensitivity in women with GDM, but in contrary gives a higher glycemic variability.

## 1. Introduction

Pregnant women develop decreased insulin sensitivity with increasing gestational age. The adapted insulin sensitivity in the mother ensures sufficient nutrients supply for the growing fetus [1]. In women with Gestational diabetes mellitus (GDM) insulin sensitivity is reduced even further and GDM is defined as decreased glucose tolerance developed during pregnancy [2,3]. The prevalence of GDM varies worldwide, ranging from 1% to over 30% in some countries and the number of women diagnosed with GDM is increasing [4,5]. 

With repeated episodes of hyperglycemia, the fetus receives too much glucose [4,6] and several studies on GDM patients have shown a correlation between increasing blood glucose levels and birth complications [7,8,9]. According to studies including pregnant women with type 1 diabetes and type 2 diabetes, large variations in blood glucose levels cause more complications than constantly elevated blood glucose levels [10,11]. Twenty-four hours continuous glucose measurements (CGM) detect a more detailed glycemic profile than self-monitored blood glucose (SMGB) by better measuring the duration and magnitude of fluctuation, especially for fasting and postprandial measurements [12,13]. 

Carbohydrate is the macronutrient with the greatest impact on postprandial blood glucose response. Treatment of GDM involves dietary guidance on the amount, type and distribution of carbohydrate [14,15,16]. However, there is currently no evidence on how carbohydrate intake should be distributed during the day. Distributing carbohydrate intake throughout the day into multiple meals and snacks may be beneficial for controlling postprandial blood glucose levels [14]. It is customary in Denmark to recommend 5–6 daily meals for patients with GDM and a maximum of 30 g of carbohydrates at breakfast [17]. However, only few intervention studies have been conducted focusing on meal pattern [14], including the optimal timing of carbohydrate intake during the day. Patients with type 2 diabetes who consume breakfast with a high energy and carbohydrate content and a low energy and carbohydrate intake in the evening showed a beneficial effect on the postprandial blood glucose response and led to an increase in Glucagon-like-peptide-1 (GLP-1), insulin response and C-peptide excretion compared to an energy and carbohydrate-reduced breakfast [18]. These findings contradict the recommendation that GDM patients should consume only a few carbohydrates for breakfast. This recommendation is given because of the usually marked postprandial rise in blood glucose during mornings, with insulin resistance being most pronounced at this time of day [19]. High carbohydrate breakfast and carbohydrate-reduced dinner could also be beneficial for blood glucose response in GDM patients. As far as the authors are aware, it has not yet been studied whether a high carbohydrate intake in the morning is beneficial in patients with GDM. 

The aim of the present study was to investigate the effects of a High-carbohydrate-morning-intake (HCM) compared to a low-carbohydrate-morning-intake (LCM), both diets isocaloric for each participant and with the same total carbohydrate content, on glycemic variability in GDM patients measured by mean amplitude of glucose excursions (MAGE) and coefficient of variation (CV), using CGM. Secondarily we investigated the effect of HCM compared with LCM on parameters of glycemic control; mean glucose (MG) and fasting blood glucose (FBG). In addition, we assessed the effect of the two diets on insulin resistance expressed as Homeostatic Model Assessment for Insulin Resistance (HOMA-IR).

## 2. Materials and Methods

### 2.1. Study Design and Population

We conducted a randomized crossover clinical trial to investigate the effect of the carbohydrate distribution on different blood glucose measurements. The study participants included women of at least 18 years of age with diet treated GDM and gestational age of at least 30 weeks. Participants were randomized through the web-based application Research Electronic Data Capture (RedCap) [20]. An independent administrative employee created a randomization-key, balanced 1:1, blinding the investigators of the allocation sequence of participants until assignment. 

Fifteen participants with GDM according to WHO diagnostic criteria (75 g, 2-h OGTT ≥ 8.5 mmol/L) [21] were enrolled. We did not include women on any type of diabetes medication, and if a participant was prescribed insulin treatment during the intervention period, the woman was excluded as well. The primary outcome was to assess the overall effect on glycemic variability in both meal plans (HCM vs. LCM) expressed by MAGE. Secondarily to investigate the effect on CV%, MG, FBG and HOMA-IR. 

All clinical data were entered in RedCap. The Danish National Committee on Health Research Ethics approved the study. All participants gave their informed consent. The study was registered at ClinicalTrials.gov (NCT03835208). 

### 2.2. Intervention

All participants received two different dietary-treatments in two independent but continuous periods of 4 days each with four to five days of wash-out period in between. The dietary treatments consisted of two different meal timing schedules with either “breakfast diet” with a high **carbohydrate** and energy content in the morning and a low carbohydrate and energy content in the evening (HCM) or a “dinner diet” with a low carbohydrate and energy content in the morning and a high carbohydrate and energy content at dinner (LCM). The total macronutrient content and composition was identical in the two periods but the meal distribution differed.

The participants visited the clinic four times during the two intervention periods; at the start and end of each intervention. Meal plans with different carbohydrate distribution was handed out followed by careful instructions from the investigators at the beginning of each intervention period. The meal plans were followed in home settings. The participants were provided with a grocery list and pictures of specific food items for shopping and the meal plans contained pictures of each meal in order to control the dietary intake as much as possible without handing out any food. 

In general, the meal plans followed the standard care of the Department of Obstetrics and Gynecology Aarhus University Hospital, apart from the carbohydrate distribution during the day. All meal plans were according to the recommendations in Table 1. All meals contained only whole grain products. The participants were asked not to ingest or drink anything besides the food components of the meal plan. All participants received a kitchen scale and were asked to weigh all foods. 

Calorie content of both the HCM and LCM meal plans were either 1800, 2000, 2200 or 2400 kcal according to the individual needs. Calorie-needs were calculated by the use of equations calculating for resting energy expenditure (REE) by Henry [22] based on pre-pregnancy weight and height multiplied by a factor of physical activity level (PAL). All participants were in their third trimester and an additional 537 kcal were added. The estimated calorie-needs were reduced by 30% if the patient was overweight (based on pre-gestational BMI >25) or already had reached the optimal weight gain, according to the recommendation of IOM [23] at the time for study inclusion. After calculating each participant’s individual energy-needs, the nearest 200-calorie step was used for guidance purposes. The two meal plans representing the two intervention periods (HCM and LCM) were kept isocaloric +/−2,5% for each participant. Both meal plans provided a total of five meals a day and had the same overall macronutrients distribution; 46 E% carbohydrates, 34 E% fat and 20 E% protein. All values +/−2 E%. 

The distribution of energy and carbohydrate content during the day for the meal plans in both intervention periods (HCM and LCM) are shown in Table 2. During both intervention-periods two 24-h diet recalls were obtained by one of the investigators to check for compliance and the participants were also asked to take pictures of the main meals and of the plate if any leftovers.

### 2.3. CGM-Measurements and -Parameter Calculation

Glucose levels were monitored continuously in all participants by CGMs (Medtronic, Ipro 2 (Medtronic MiniMed, Northridge, CA, USA)) for 72 h in each intervention period. On day 1 of each of the intervention periods, the sensor was attached to the triceps of the participant by one of the investigators immediately before serving breakfast. The sensor was affixed with an adhesive bandage and removed at the end of each intervention period by an investigator. The women were instructed to do pre-prandial SMBG four times a day (before breakfast, lunch, dinner and before bedtime) with a glucometer (Accu-Chek Mobile, Roche Diagnostics GmbH, Mannheim, Germany) for calibrating the CGMs. Accuracy of the glucometer was tested in comparison with the fasting blood glucose tested in the laboratory and a 15% deviation was accepted. 

Calibrated glucose profiles recorded for each patient over a period of 48 h of each intervention period (day 2 and day 3) was used for statistical analysis and quantification of glycemic variability. Parameters of glycemic variability included the standard deviation of blood glucose (SD), MAGE and the CV%. MG was also calculated from the CGM data as a measurement of glycemic control. 

### 2.4. Blood Samples

Fasting blood samples were drawn at all four visits. C-peptide, fasting blood glucose, lipid-profile (triglycerides, HDL, LDL and total cholesterol), 3-hydroxy-butyrat and C-reactive protein (CRP) were measured. All blood samples were drawn at the same time of the day at all four visits and analyzed by a certified medical laboratory technician immediately after sampling.

HOMA-IR was calculated based on fasting C-peptide and fasting blood glucose at baseline and at the end of each intervention period using the HOMA Calculator v2.2.3 [24].

### 2.5. Sample Size and Power Analysis

Power calculation on the primary outcome MAGE was based on a study by Dalfrá et al. [25]. A sample size of 12 participants was required to ensure adequate power of 80%, to detect a 5% difference between interventions. With an estimated dropout range of 20%, it was necessary to include a total of 15 participants. 

### 2.6. Statistical Analysis

Data of baseline characteristics are presented by descriptive statistics for all included participants (*n* = 12). The efficacy of the two interventions was assessed based on the within-subject difference between the two meal schedules regarding the outcome variable. Outcome from CGM data are calculated by the use of the excel-based workbook software called “EasyGV” [26]. 

To ensure that treatment effects were distinguished from carry-over effects, test of carry-over effect was done by performing a pre-test for unpaired samples. An unpaired two sample *t*-test was used. 

Comparison of treatment effects of HCM and LCM was conducted using paired sample *t*-tests on the difference between periods. Assumptions of equal mean and SD was tested using Bland-Altman plots. Whether or not outcome differences were normally distributed were assessed with QQ-plots. A *p*-value ≤ 0.05 was considered statistically significant. Statistical analysis was performed using R-statistics software.

All data are represented as mean ± standard deviation (SD).

## 3. Results

### 3.1. Participants

A total of 12 out of 15 participants completed the study. Two of the excluded participants were randomized to sequence LCM|HCM and one to sequence HCM|LCM, leaving a total of six participants in each sequence-group for the final analysis. A flow chart of all study participants is shown in Figure 1.

Baseline characteristics of the study population are shown in Table 3. Study population had a mean gestational age (GA) of 33.5 ± 2.3 weeks at the beginning of the intervention, a mean age of 33.6 ± 6.7 years and a mean BMI of 25.2 ± 4.0 kg/m^2^.

### 3.2. Comparison between HCM and LCM-Diet on Parameters of Glycemic Variability and Glycemic Control

Table 4 shows the mean values of MAGE, CV, SD, TIR, TBR, TAR and MG of both HCM and LCM and the difference in C-peptide and FBG of the two attendance days in each period (day 4 minus day 1). Statistically significant estimates of the treatment effect from within-subject differences (HCM minus LCM) were observed for measurements of glycemic variability (MAGE and CV) and glycemic control (MG and FBG). 

Both measurements of glycemic variability indicated a significantly higher variability in the HCM-diet compared to the LCM diet. MAGE was significantly higher on the HCM-diet (2.5 ± 1.8 mmol/L) compared to the LCM-diet (1.9 ± 0.5 mmol/L) with a mean difference of 0.7 mmol/L (0.3; 1.2) (*p* = 0.004) (Table 4 and Figure 2). A higher CV was found with a mean difference of 5.1% (1.5; 8.8) (*p* = 0.01) on the HCM-diet compared to the LCM-diet. 

Both measurements of glycemic control were significantly higher for LCM- compared with HCM-diet. A treatment effect with a mean difference of −0.3 mmol/L (−0.6; 0.1) (*p* = 0.02) in MG was observed when comparing HCM-diet to LCM-diet (Table 4 and Figure 3). A decrease in FBG was observed for the HCM-diet with a mean FBG at baseline of 4.9 mmol/L and 4.6 mmol/L at the end (mean difference of −0.2 mmol/L), while an increase was found for the LCM-diet with a mean FBG at baseline of 4.9 mmol/L and 5.1 mmol/L at the end (mean difference of 0.2 mmol/L). There was a significant difference of FBG between the two diets of −0.4 mmol/L (−0.7; −0.1) (*p* = 0.01) when comparing the mean difference (day 4 minus day 1) of HCM- to LCM-diet. C-peptide did also tend to decrease during HCM-diet with −82.3 pmol/L, while increasing during LCM-diet with 71.9 pmol/L between end of each period and baseline.

In summary, the HCM-diet seemed to give a higher glycemic variability but a better glycemic control when comparing with the LCM-diet. 

### 3.3. HOMA-IR

To assess insulin resistance HOMA-IR was calculated based on fasting C-peptide and FBG at baseline and at the end of each intervention. HOMA-IR decreased significantly during the HCM-period with a mean difference of −0.214 (*p* = 0.02) and tended to increase during the LCM-period with a mean difference of −0.107 (*p* = 0.68). However, there was not a significant difference when comparing the HCM- with the LCM-period with a mean difference of −0.321. 

### 3.4. Tertiary Outcomes

Table 5 shows the mean values of the differences (day 4 minus day 1) between the two interventions, regarding the lipid-profile, 3-hydroxy-butyrat and C-reactive protein. Comparisons between the two diets by a paired *t*-test did not reveal any significant within-subject differences (*p* > 0.05 for all). 

### 3.5. Food Intake

Table 6 shows the comparison of food intake in the two diets based on the mean of the two 24-h recalls. 

Some intake of the participants deviated by more than the acceptable (2.5% from intended energy content (approximately +/−50 kcal) and +/−2 E% for macronutrient). Even though there were no significant difference within-subject in intake of energy, carbohydrate and protein when comparing the two intervention periods (HCM vs. LCM) (Table 6), there was a significant difference in the intake of fat between the two periods with a mean difference of 9.9 g. 

The difference in fat intake between the two intervention diets might have been due to misunderstandings. One participant thought that she was supposed to put butter on every piece of bread in the LCM intervention and some participants used recipes for wok (also in the LCM intervention) in which they did not use light coconut milk as prescribed, but used regular coconut milk instead.

## 4. Discussion

### 4.1. Glycemic Variability and Glycemic Control

This study showed that a HCM diet (with a high carbohydrate intake in the morning) gave a higher MAGE and CV%, indicating a higher glycemic variability, when comparing with a LCM diet (with a low carbohydrate intake in the morning and a higher carbohydrate intake in the evening). On the other hand, the HCM diet seemed to lower mean glycemia compared to LCM-diet; MG was significantly lower in the HCM-diet and likewise was FBG. Similar findings have to our knowledge not been shown before. However, Jakubowicz et al. reported that morning carbohydrate intake compared to low carbohydrate intake lowered postprandial hyperglycemia in non-pregnant type 2 diabetic patients [18]. Though, comparison of results between the two studies are complicated, because our study evaluated treatment effect over a period, whereas Jakubowicz evaluated a single meal response, our results of glycemic control expressed as MG and FBG, seem to be in accordance with the results of Jakubowicz; a high carbohydrate intake in the morning seems beneficial. The results of glycemic variability are on the other hand not in accordance with the study by Jakubowicz as these parameters seem to be in favor of the LCM-diet in our study. 

Large variations in blood glucose levels may cause more complications than constantly elevated glucose levels in patients with type 1 diabetes and type 2 diabetes [10]. Limited evidence are available for the consequence of high glycemic variability in pregnancies complicated by gestational diabetes [25]. It is evident that hyperglycemia, one characteristic of higher variability, is a strong predictor of adverse pregnancy outcomes, especially LGA [7]. However, in a new study from 2018 in GDM patients, glycemic variability in the third trimester was not a predictor of fetal birthweight and had no significant association to adverse pregnancy outcomes [27]. In the present study all participants had a stable glucose values defined as CV <36% [28], why one could argue that the difference in glycemic variability between the diets only have a small impact. We also found a mean difference of 0.7 mmol/L for MAGE, and it is debatable if this has any clinical relevance. In addition, TIR was highest on the LCM diet compared to HCM diet, primarily due to a decrease in TBR supporting the LCM diet as most favorable. In an ATTD consensus report from 2019 on clinical CGM targets of the metrics TIR, TBR and TAR, consensus was reached on glycemic cut point: Target range 3.5–7.8 mmol/L in pregnancy along with a target for time pr. day of 70% in late gestation (>34 weeks). No significant differences were observed in TIR between groups and despite diet the average TIR was above the recommended 70%. Whether glycemic control or variability is the best measurement to predict complications in diabetic pregnancies is also debatable, but ongoing studies will contribute to a clarification. 

### 4.2. Carbohydrate Content of the Two Diets

In this crossover study, it can be argued that the LCM-diet is comparable to standard care at Department of Obstetrics and Gynecology at Aarhus University Hospital, as the carbohydrate content of the LCM-diet was approximate similar to the 30 g of carbohydrate usually recommended in standard care. In this way, the study contributes with new knowledge that contradicts the recommendations used in practice. 

The study has a number of strengths and limitations. First, the crossover design made it possible to use within subject analyzes thus reducing the risk of potential confounders. The calorie-needs were individually calculated which can be seen as a strength as long as within-subject analyzes are used. 

It is important that the total carbohydrate intake is the same during the two periods and this was the case in our study. According to the 24-h recall the total carbohydrate content did not differ significantly between the two interventions (Table 6). There was however a significant difference in the total intake of fat between the two groups. The higher intake of fat might have interfered with some of the outcomes, but because carbohydrate is the macronutrient with the highest impact on the blood glucose, we do not expect that this has affected the results noticeably. 

Two be able to compare the two diets the breakfast meal of the HCM diet had the same carbohydrate content +/−2 E% as the dinner meal of the LCM diet. The dinner usually contains a higher carbohydrate load than breakfast. In the matter of transferring the results to clinical practice, this therefore needs to be taken into account. However, the participants in our study all reported satiety after dinner and some even had trouble finishing the whole meal even though most of the participants were energy reduced due to either BMI ≥25 or large weight gain at enrollment. 

Clinical guidelines recommend a daily meal frequency of 3 main meals and 2–3 snack meals in between. This is advised in order to avoid high energy intake at main meals and distribute the carbohydrate intake over the day, and thereby decreasing the risk of postprandial hyperglycemia. Both diets in our study had a meal frequency of 5 meals pr. day. The HCM diet had a snack meal between breakfast and lunch, but no late evening snack and the LCM diet had no snack between breakfast and lunch, but a late evening snack. 

### 4.3. CGM Data

Verification of CGM data was necessary to make sure that data on glycemic patterns was reliable and representative. No standard is currently available for CGM validation and performance, but the ATTD consensus recommendations are used as the best alternative [28]. 

The CGM system was calibrated based on measurements of SMBG with a personal glucometer (Accu-chek mobile). According to our analyzes the accuracy of the glucometer was within a range of +/−15% as advised by the ISO standard and ADA. Furthermore, the CGM IPro has been validated and used in pregnant women before [29,30]. 

Instructions were given to do pre-prandial SMBG four times a day. Not all participants managed to do all instructed measurements which weakens the calibration of the CGM system and implies a weakness in data reliability when dependent on compliance. The ATTD consensus report also recommends a minimum of 2 weeks to secure a sufficient data collection period. Our results were only based on data of 2 full days (2 × 24 h) of each intervention period. However, we did find significant differences on the parameters MAGE, CV%, FBG and MG between the two diets. The differences might have been even greater if we had a larger data set. 

Another limitation of this study is the small sample size. However, the power was fulfilled, and it would be unethical to include more patients, and despite a small dataset, we were able to detect significant differences on several parameters between the two diets. 

## 5. Conclusions

In conclusion, a diet with a high carbohydrate intake in the morning seem to result in a higher glycemic variability, but a lower MG and FBG, when comparing with a diet with a low carbohydrate intake in the morning and a higher carbohydrate intake in the evening. In addition, insulin resistance expressed as HOMA-IR decreased significantly during the HCM diet, indicating that a carbohydrate distribution of 50% in the morning favors lower blood glucose and improvement in insulin sensitivity in women with GDM.

Further research may clarify the impact of carbohydrate distribution on the complex matter of blood sugar regulation, and the use of CGM data to quantify glycemic variability, glucose control and their interactions. 

## Figures and Tables

**Figure 1 nutrients-12-00475-f001:**
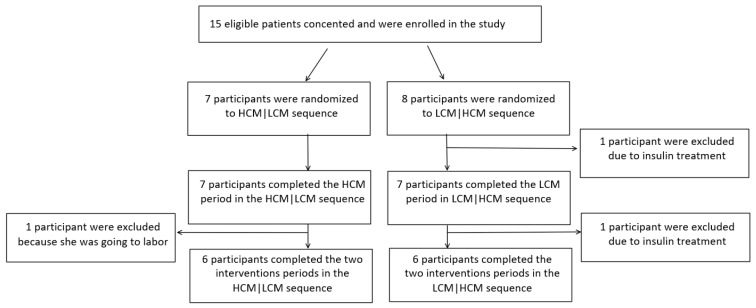
Flow chart of the study. HCM, high carbohydrate and energy content in the morning; LCM, low carbohydrate and energy content in the morning.

**Figure 2 nutrients-12-00475-f002:**
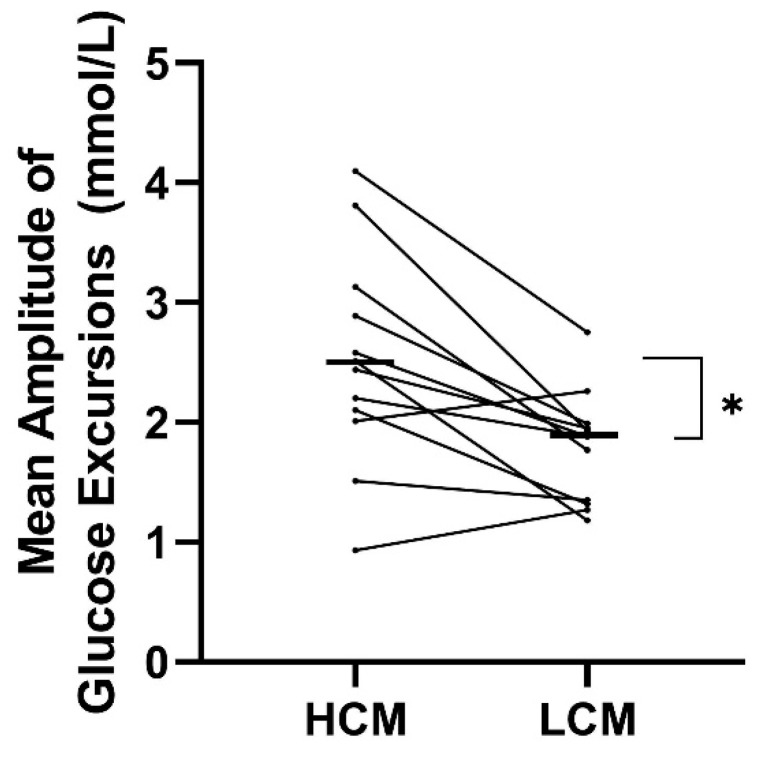
Treatment effect on mean amplitude of glucose excursions (MAGE) for each participant following a diet with two days of high carbohydrate load (HCM) and two days of low carbohydrate load (LCM) in the morning. Mean values of the treatment effect of each period are marked with black lines. HCM, high carbohydrate and energy content in the morning; LCM, low carbohydrate and energy content in the morning; MAGE, mean amplitude of glucose excursions.

**Figure 3 nutrients-12-00475-f003:**
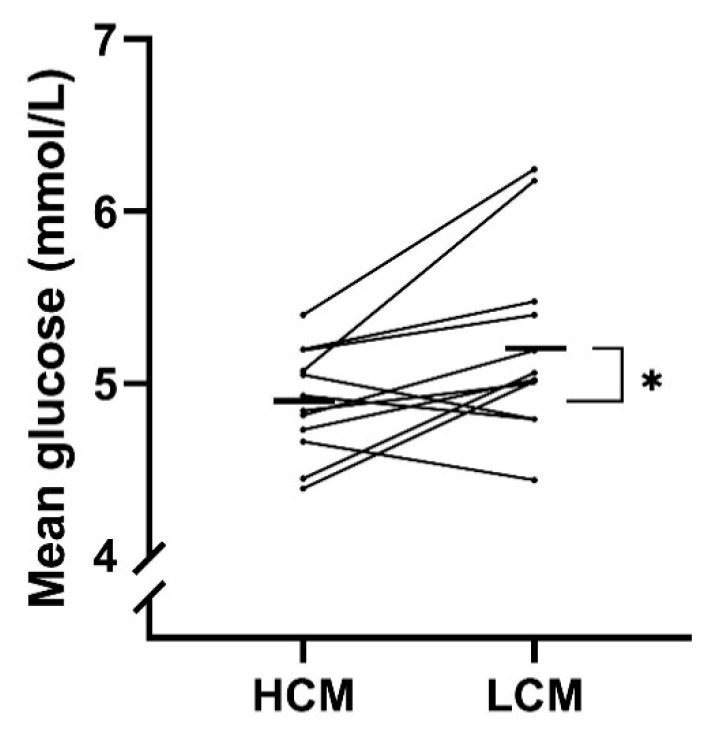
Treatment effect on mean glucose (MG) for each participant following a diet with high carbohydrate load (HCM) and a diet with low carbohydrate load (LCM) in the morning. Mean values of the treatment effect of each period are marked with black lines. HCM, high carbohydrate and energy content in the morning; LCM, low carbohydrate and energy content in the morning; MG, mean glucose.

**Table 1 nutrients-12-00475-t001:** Meal plans recommendations.

Nutrient	
Carbohydrates	45–60 E%
Fibers	>28 g
Sugar (added)	10 E%
Protein	10–20 E%
Fat	25–40 E%
SFA	<10 E%
PUFA	5–10 E%
MUFA	10–20 E%

SFA, saturated fatty acid; PUFA, polyunsaturated fatty acid; MUFA, monounsaturated fatty acid.

**Table 2 nutrients-12-00475-t002:** Meal plan structure and diet compositions of HCM and LCM.

	HCM	LCM
Energy distribution Calories as a percentage of total calorie content during the day	Breakfast: 25%–30%	Breakfast: 15%–20%
	morning-snack: 15%–20%	Lunch: 25%–30%
	Lunch: 25%–30%	Afternoon-snack: 10%–15%
	Afternoon-snack: 10%–15%	Dinner: 30%–35%
	Dinner: 15%–20%	Late-night-snack: 15%–20%
Carbohydrate distribution Carbohydrate as a percentage of total carbohydrate content during the day	morning: 50% Breakfast: 30%–35% morning-snack: 15%–20% Lunch: 40% Lunch: 25%–30% Afternoon-snack: 10%–15% Dinner: 10%	morning: 10% Lunch: 40% Lunch: 25%–30% Afternoon-snack: 10%–15% Dinner: 50% Dinner: 30%–35% Late-night-snack: 15%–20%

HCM, high carbohydrate and energy content in the morning; LCM, low carbohydrate and energy content in the morning.

**Table 3 nutrients-12-00475-t003:** Baseline characteristics of the participants (*n* = 12) included in the final analysis.

	All Study Participants (*n* = 12)
Age at Debut (year)	33.6 (6.7)
Pregestational weight (kg)	68.6 (11.3)
GA (weeks)	33.5 (2.3)
Parity (n (%))	
0	6 (50)
1	2 (17)
2	4 (33)
BMI (kg/m^2^)	25.2 (4.0)
GWG (kg)	12.8 (7.7)
OGTT (mmol/L)	9.7 (0.7)
HbA1C	5.3 (2.5)
(%) (mmol/mol)	(34.4 (4.2))
Average BS (mmol/mol)	5.8 (0.6)
Systolic BP (mmHg)	115.1 (9.4)
Diastolic BP (mmHg)	75.4 (6.3)

Results are mean (SD) except parity which is n (%). GA, gestational age; GWG, gestational weight gain.

**Table 4 nutrients-12-00475-t004:** Results of treatment effect.

	HCM (*n* = 12) Mean (SD)	LCM (*n* = 12) Mean (SD)	Difference (95% CI)	*p*-Value
**Glycemic Variability**				
MAGE (mmol/L)	2.5 (1.8)	1.9 (0.5)	0.7 (0.3;1.2)	0.004
CV (%)	20.1 (5.9)	14.9 (3.6)	5.1 (1.5;8.8)	0.01
SD	1.0 (0.3	0.8 (0.2)	0.2 (0.0;0.4)	0.02
TIR (%)	93.46(8.7)	97.96(3.2)	−4.5(−9.7;0.7)	0.08
TBR (%)	6.42(8.5)	2.04(3.2)	4.38(-0.7-9.5)	0.09
TAR (%)	1.64(2.6)	1.06(2.5)	0.58(-0.78;1.93)	0.37
**Glycemic Control**				
MG (mmol/L)	4.9 (0.3)	5.2 (0.5)	−0.3 (−0.6; −0.1)	0.02
∆C-peptide (pmol/L)	−82.3 (109.1)	71.9 (363.9)	−154.2 (−381.4;73.0)	0.16
FBG_start_	4.85(0.5)	4.88(0.6)	−0.025(−0.2;0.1)	0.75
FBG_end_	4.62(0.4)	5.07(0.5)	−0.45(−0.7; −02)	0.0007
∆FBG (mmol/L)	−0.2 (0.2)	0.2 (0.5)	−0.4 (−0.7; −0.1)	0.01

All variables are given as mean (SD). HCM, high carbohydrate and energy content in the morning; LCM, low carbohydrate and energy content in the morning; MAGE, mean amplitude of glucose excursions; CV, coefficient of variation; TIR, time in range; TBR, time below range; TAR, time abover; MG, mean glucose; FBG, fasting blood glucose. All FBG values are taking in the morning. FBG_start_ is day 1 in each period. FBG_end_ is day 4 in each period. Values of MAGE, CV and MG are all based on calculation of the calibrated glucose profiles recorded for each patient over a period of 48 h of each intervention period (day 2 and day 3). ∆: Delta values are based on day 4 minus day 1 in both HCM and LCM intervention period.

**Table 5 nutrients-12-00475-t005:** Effect on tertiary outcomes.

	HCM (*n* = 12) Mean (SD)	LCM (*n* = 12) Mean (SD)	Difference (95% CI)	*p*-Value
∆p-total cholesterol (mmol/L)	0.0 (0.250)	0.1 (0.2)	−0.0 (−0.2; 0.2)	0.87
∆p-LDL cholesterol (mmol/L)	−0.2 (0.408)	0.2 (0.4)	−0.4 (−0.9; 0.0)	0.07
∆p-HDL cholesterol (mmol/L)	0.0 (0.1)	0.1 (0.1)	−0.0 (−0.1; 0.1)	0.83
∆p-triglycerides (mmol/L)	−0.0 (0.3)	−0.2 (0.5)	0.3 (−0.1; 0.6)	0.15
∆CRP (mg/L)	−0.8 (2.1)	1.4 (4.5)	−2.2 (−6.3; 1.9)	0.27
∆3-hydroxy-byturat (mmol/L)	−0.2 (0.8)	−0.5 (1.0)	0.2 (−0.7; 1.1)	0.57

HCM, high carbohydrate and energy content in the morning; LCM, low carbohydrate and energy content in the morning. *p*-values and 95% CI are based on paired *t*-tests with an assumption of equal variance (all except CRP) and approximately normally distributed data. ∆: Delta values are based on day 4 minus day 1 in both HCM and LCM intervention period.

**Table 6 nutrients-12-00475-t006:** Intra-individual comparison of food intake between HCM and LCM.

	HCM Mean (SD)	LCM Mean (SD)	Difference (95% CI)	*p*-Value
Energy, kcal	2012 (263)	2055 (2740)	−43.33 (−126.6; 40.0)	*p* = 0.28
Carbohydrates, g	222 (28)	215 (36)	6.2 g (−2.9; 15.4)	*p* = 0.16
Dietary fiber, g	38.79 (5.8)	39.50 (8.0)	−0.71(4.3; 2.9)	*p* = 0.68
Fat, g	73 (10)	82 (10)	−9.9 g (−16.5; −3.2)	*p* = 0.007
Protein, g	98 (15)	94 (14)	4.2 g (−0.6; 8.9)	*p* = 0.08

HCM, high carbohydrate and energy content in the morning; LCM, low carbohydrate and energy content in the morning. Mean difference, 95% CI and *p*-values are based on a paired *t*-test between HCM and LCM based on raw data.

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
