# Peer review of "Effect of High Versus Low Carbohydrate Intake in the Morning on Glycemic Variability and Glycemic Control Measured by Continuous Blood Glucose Monitoring in Women with Gestational Diabetes Mellitus—A Randomized Crossover Study"

_nutrients, 2020, doi:10.3390/nu12020475_

Round 1

Reviewer 1 Report

This study addressed two different distribution pattern of energy and available carbohydrate during the day with either a high load during the morning in two occasion (breakfast and snack) and a low load at dinner compared to a low load for breakfast and a high load in the evening provided in two occasions: dinner and late snack in women with gestational diabetes. The studied markers were the calculated parameters obtained using the continuous glucose monitoring. Additional blood markers (C-peptide, fasting blood glucose, lipid profile) were evaluated at fasting state before and after the intervention. 15 subjects were included in a cross over design. The analysis was done on 12 pregnant women.
The authors found interesting results of the glycemic variability in favor of the pattern of high load of energy and available carbohydrates during evening period compared to a high load during morning period. They found as well interesting result of the Mean Glucose and Fasting Blood glucose which showed lower values after the high load in the morning compared to the evening.

The authors were able to select and adjust the pregnant women at the same gestational stage. They were able to evaluate the effect of the diet out of medication interference. In addition, the energy intake was adjusted with the expected energy expenditure.

Major remarks:

The importance of the diet control in this study is crucial to better understand the link between the ingested foods and the glycemic control in gestational diabetes.
In line 105 – 106, “The total macronutrient content and composition was identical in the two periods but the meal distribution differed. “
In line 108 – 109: Meal plans with different carbohydrate distribution was handed out followed by careful instructions from the investigators at the beginning of each intervention period. The meal plans were followed in home settings”

1) Could you clarify if you provided the foods to the subjects? If yes, could you precise the menus you provided? If no, could you precise the meal recommendations you gave?

2) Regarding the comparison of daily diet pattern, there is a major piece of the description which is missing: the carbohydrate digestibility: there is no information regarding slowly of rapidly digestible carbohydrates
The carbohydrate digestibility interfere dramatically with the postprandial glycemic regulation. Indeed, the metabolic profile can be different in healthy or insulin resistant subjects or type 2 diabetes patients depending on the digestibility profile of the available carbohydrates (Seal et al. 2000; Harbis et al. 2004, Vinoy et al. 2013; ADA 2019). Without the control of these parameters, no data interpretation can be done regarding the dietary profile.

3) The authors mentioned carbohydrates during all the manuscript. I assume that they addressed the information regarding available carbohydrates. If yes, please precise.
In addition , there is no information about:
- There is no information regarding non available carbohydrates as resistant starch
- There is no information regarding fiber intake
These complementary components interfere in glycemic response and need to be controlled in both arms of the design.

4) In the results part, the figure 4 does not bring additional information and should be removed.

5) In figure 3, I did not find the information regarding the mean of “Period Effect”. Could you precise in the legend of the figure and in the text, please

6) In the discussion part, the authors confirmed that a pattern of high carbohydrate and energy load at breakfast is detrimental for glycemic variability determined by MAGE, CV and SD, compared to a low load. These markers have been associated to positive effect of HBA1c in several papers (Ohara et al. 2016; Caprnda et al. 2017; Chan CL et al. 2017; Li et al. 2017).
In another hand, you mentioned that with the high carbohydrate and energy load during evening, there is an improvement of Mean glycemia and fasting blood glucose, compared to low load. These results should be completed by additional parameter provided with continuous glucose monitoring which is the Time In Range (TIR) during the postprandial periods. This information should help to better describe the glycemic profiles and help to determine if in this case, the mean value or the variability is more important in gestational diabetes regulation

7) In the discussion part, the different dietary patterns and the associated timing were not discussed. The snack moments were not provided at the same time, which modified the postprandial regulation during the morning and late at the evening. Please add a part with the potential effect on the glycemic control

8) The hypothesis of having the most important intake of available carbohydrate at lunch has not been discussed. Why?

Minor comments
1) Line 21: please correct “Correspondence:orresponding author”
2) In the material and methods, the clear duration of each period is not precise. In the results section, there is the information of 4 days of intervention. Could you clarify it in the mat&meth section please?
3) Please clarify:
Line 271: “the HCM diet seemed to be in favor of the LCM-diet when looking at variables of glycemic control;”

Reviewer 2 Report

Obtained results are very interesting however, they are controversial. Therefore these studies need further investigations according to Authro's suggestions: Lines 299-307 and very important: line 323 (small sample size).  

Reviewer 3 Report

Rasmussen and colleagues explored the impact of carbohydrate intake distribution in women with gestational diabetes. The use of continuous glucose monitoring (CGM) provided an opportunity to look in more detail into the relation of meal ingestion and glucose excursions. The study design was based on a crossover between high/low carbohydrate intake at breakfast/dinner, maintaining isocaloric daily intake. Although the sample was small, the authors declared to be within calculated power analysis for the primary outcome: MAGE.

The subject of the manuscript has a direct clinical impact, as dietary guidance is a major component of the support to patients with gestational diabetes. However, by choosing the present study design, the authors ignored the influence of dinner composition, and especially carbohydrate load, in the response to breakfast. Thus, discussion of results should consider the effect of dinner high/low carbohydrate intake.

Regarding the description of methodology, I was not able to clearly identify the period length for each diet exposure. Furthermore, data for blood parameters are given as deltas, when the absolute values are perhaps even more of interest. Another weaker point is that only 2 days of CGM data were considered.

Round 2

Reviewer 3 Report

I thank the authors for the clarifications. I still believe that the study design hinders the draw of conclusions as stated by the authors, and should be reconsidered. Furthermore, the sample is small and the length of observation considerably narrow.